# Heterostructured NiO/ZnO Nanorod Arrays with Significantly Enhanced H_2_S Sensing Performance

**DOI:** 10.3390/nano9060900

**Published:** 2019-06-20

**Authors:** Dongyi Ao, Zhijie Li, Yongqing Fu, Yongliang Tang, Shengnan Yan, Xiaotao Zu

**Affiliations:** 1School of Physical, University of Electronic Science and Technology of China, Chengdu 610054, China; aodongyi@std.uestc.edu.cn (D.A.); zhijieli@uestc.edu.cn (Z.L.); shengnanyan_uestc@163.com (S.Y.); 2Faculty of Engineering and Environment, Northumbria University, Newcastle upon Tyne NE1 8ST, UK; richard.fu@northumbria.ac.uk; 3School of Physical Science and Technology, Southwest Jiaotong University, Chengdu 610031, China; tyl@swjtu.edu.cn

**Keywords:** ZnO, nanorods, NiO, p-n junction, H_2_S gas sensor

## Abstract

H_2_S gas sensors were fabricated using p-n heterojunctions of NiO/ZnO, in which the ZnO nanorod arrays were wrapped with NiO nanosheets via a hydrothermal synthesis method. When the H_2_S gas molecules were adsorbed and then oxidized on the ZnO surfaces, the free electrons were released. The increase in the electron concentration on the ZnO boosts the transport speed of the electrons on both sides of the NiO/ZnO p-n junction, which significantly improved the sensing performance and selectivity for H_2_S detection, if compared with sensors using the pure ZnO nanorod arrays. The response to 20 ppm of H_2_S was 21.3 at 160 °C for the heterostructured NiO/ZnO sensor, and the limit of detection was 0.1 ppm. We found that when the sensor was exposed to H_2_S at an operating temperature below 160 °C, the resistance of the sensor significantly decreased, indicating its n-type semiconductor nature, whereas when the operating temperature was above 160 °C, the resistance significantly increased, indicating its p-type semiconductor nature. The sensing mechanism of the NiO/ZnO heterostructured H_2_S gas sensor was discussed in detail.

## 1. Introduction

Nanowires [1,2,3], nanorods (NRs) [4,5], nanobelts [6,7], and nanotubes [8] have shown good electronic or optoelectronic properties due to their special nanostructures, excellent crystallinities and oriented growth, significant quantum size effect, and optical non-linearity [9]. Among these, zinc oxide (ZnO) NRs are widely applied in the fields of sensors [4,10,11,12,13,14], photocatalysis [13], solar cells [15,16], UV photodetectors [17,18], and field emission devices [19] due to their wide band gap (3.37 eV), high binding exciton energy (60 meV), excellent electricity properties, and low cost [20,21]. Compared with bulk ZnO materials, ZnO nanorod arrays have a large specific surface area and strong electronic transmission capability [22].

Hydrogen sulfide (H_2_S), one of the most common toxic gases, causes serious issues in terms of human health and environmental safety [23,24]. The H_2_S concentration that is safe for human beings is about 20–100 ppb [24]. The U.S. Scientific Advisory Board on Toxic Air Pollutants recommends that the concentration of H_2_S in human environments should be controlled within <83 ppb [25]. Therefore, it is critical to develop a fast, precise, and reliable H_2_S detection method [23]. In recent years, the H_2_S gas sensor has received considerable attention for its high sensitivity, good selectivity, good reliability, low cost, and low power consumption. Many H_2_S gas sensors based on Fe_2_O_3_, CuO, ZnO, and CeO_2_ semiconductors have been reported [3,4,26,27,28,29,30,31]. Among these metal oxides, ZnO nano-material is among the most studied. However, the sensing performance of the nanostructured sensors is strongly dependent on the morphology and structure of sensitive materials. To improve the performance of these nanostructured sensors, an effective method involves using surface modification. For example, Zhao et al. [32] reported that a gas sensor composed of ZnO nanofibers modified with Cu can achieve a response of 18.7 to 10 ppm H_2_S at 230 °C. Compound semiconductor oxides, which can form p-n heterostructures, have also been employed extensively to improve H_2_S sensors’ properties. For example, Kim et al. [33] reported that a gas sensor composed of ZnO NRs modified with CuO showed a response of 890 when exposed to 50 ppm H_2_S at 500 °C. Nickel oxide (NiO), as a typical p-type semiconductor [14], has been extensively researched in optoelectronic devices and sensors due to its ease of formation of a p-n heterojunction together with an n-type semiconductor (such as ZnO), thus achieving excellent physical and chemical properties [18,34,35]. However, few studies of NiO/ZnO p-n heterojunciton nanorod arrays have been reported for detecting H_2_S gas.

In this paper, a p-n heterojunction structure composed of NiO nanosheets (NSs) and ZnO NRs was prepared using a hydrothermal synthesis method. It was used as a sensitive material to fabricate a gas sensor on an Al_2_O_3_ ceramic tube. Excellent H_2_S sensing performance was attained, and the sensing mechanism was discussed in detail.

## 2. Experiment

All the chemicals used were analytical grade reagents and purchased from Sinopharm Chemical Reagent Co. Ltd. (Shanghai, China). All the standard gases used were purchased from NIMTT Measurement and Testing Technology Co. Ltd. (Chengdu, China). The crystalline phase of samples was characterized by X-ray diffractometer (XRD, Dmax-2500, Rigaku, Tokyo, Japan) and transmission electron microscope (TEM, JEM-2200FS, Jeol, Tokyo, Japan). The morphologies and element contents of the samples were observed using scanning electron microscope (SEM, Inspect F50, ThermoFisher, Hillsboro, OR, USA) and energy dispersive spectrometer (EDS, Inspect F50, ThermoFisher, Hillsboro, OR, USA). UV–visible spectrophotometer (UV-2600, Shimadzu, Kyoto, Japan) was used to study the band-gap energies of the samples. X-ray photoelectron spectroscopy (XPS, XSAM800, Kratos, Manchester, UK) was measured to analyze valence states of the elements in the samples.

### 2.1. Growth of ZnO NRs

A hydrothermal synthetic method was used to grow ZnO NRs. The typical procedure was described below: 1.646 g (7.5 mmol) zinc acetate dihydrate of was dissolved into 25 mL 2-methoxyethanol, and then 0.45 mL ethanolamine was added. The above solution was stirred for 1 h at 60 °C to achieve a sol, which was then uniformly coated onto the cleaned Al_2_O_3_ tubes and annealed at 300 °C for 10 min. The process was repeated twice and annealed at 450 °C for 1 h to form ZnO seeds layer on the Al_2_O_3_ tubes. We mixed 40 mL (5 mM) zinc nitrate aqueous solution and 40 mL (5 mM) hexamethylenetetramine, and then the solution was placed in a Teflon-lined stainless steel autoclave. The Al_2_O_3_ tubes covered by the ZnO seeds layer were also placed in the autoclave. After hydrothermal reaction at 90 °C for 5 h, a ZnO NR layer formed on the Al_2_O_3_ tubes. Finally, the sample was rinsed three times in deionized water and dried in an oven at 80 °C.

### 2.2. Preparation of NiO/ZnO Heterostructures

Nickel acetate tetrahydrate (1.866 g; 7.5 mmol) and 0.45 mL ethanolamine were added into 25 mL 2-methoxyethanol, then a uniform sol was obtained by stirring for 3 h at room temperature. The sol was uniformly coated onto the ZnO NRs’ surfaces, and then annealed at 300 °C for 10 min. The process was repeated again and then annealed at 450 °C for 1 h to obtain a NiO seeds layer. Nickel nitrate (40 mL; 5 mM) and hexamethylenetetramine (40 mL; 5 mM) were mixed into a uniform solution and transferred into the Teflon-lined stainless steel autoclave, and the Al_2_O_3_ tubes with ZnO NRs were also placed in the autoclave, which was heated at 90 °C for 3 h. The Al_2_O_3_ tubes were removed and rinsed three times in deionized water and dried in the oven at 80 °C to obtain a NiO/ZnO heterostructure sample.

## 3. Results and Discussion

### 3.1. Characterizations of Pure ZnO Nanorods and NiO/ZnO Heterostructures

SEM images of pure ZnO NRs and NiO/ZnO heterostructures are shown in Figure 1. ZnO NRs are uniform with an average diameter of about 150 nm, as shown in Figure 1a. The quasi-hexagonal cross-section of ZnO NRs indicates that ZnO NRs are preferentially oriented in a direction of [1], indicating a wurtzite crystal structure. The SEM images of the NiO/ZnO heterostructures sample shown in Figure 1b,c indicate that the NiO NSs shows a netlike morphology on the top of ZnO NRs, and fills some gaps between ZnO nanorods, which facilitated the formation of p-n heterostructures at their interfaces. Figure 1d depicts the EDS spectrum of NiO/ZnO heterostructures, based on which the atomic content of nickel was estimated to be 2.41%.

XRD pattern of pure ZnO NRs is shown in Figure 2. The ZnO NRs are in a hexagonal wurtzite crystal structure with lattice constants a = 3.252 Å and c = 5.201 Å (JCPDS file No. 36-1451). The three main peaks are at (0 0 2), (1 0 1), and (1 0 3). As the (0 0 2) diffraction peak is the strongest, we conclude that ZnO NRs grow preferentially along this direction with high crystallinity. For the NiO/ZnO heterostructures, no peaks corresponding to NiO are detected and the XRD peak positions of ZnO do not shift because the quantity of NiO is less than the XRD detection limit, estimated to be at 5%.

TEM analysis was used to further investigate the NiO/ZnO heterostructures. Figure 3a–d show the TEM bright-field image of NiO/ZnO heterostructures and elemental mappings of Zn, Ni, and O. Zn is concentrated on the ZnO NRs and Ni is diffused over the interfaces of the ZnO NRs network, which means that the gaps and surfaces of ZnO NRs are covered with NiO NSs.

Figure 4a shows the survey XPS spectrum of NiO/ZnO heterostructures, which indicates the existence of Zn, Ni, and O. Figure 4b shows the high resolution Zn 2p peak, which has been deconvoluted into peaks at 1021.4 eV and 1044.6 eV corresponding to Zn 2p_3/2_ and Zn 2p_1/2_, respectively. The results reveal the existence of Zn^2+^ in ZnO [18,36]. Figure 4c shows the Ni 2p spectrum of NiO/ZnO heterostructures. The peaks at the binding energies of 855.6 eV and 873.2 eV correspond to Ni 2p_3/2_ and Ni 2p_1/2_ of NiO, respectively. Apart from these, two satellite peaks located at the binding energies of 861.5 eV and 879.3 eV reflect Ni 2p_3/2_ and Ni 2p_1/2_, respectively, which proves the existence of Ni^2+^ [18]. The O 1s XPS spectrum of the NiO/ZnO heterostructures is shown in Figure 4d. The two peaks that deconvoluted at 530.8 eV and 532.3 eV correspond to the metal oxides (Zn-O bonds and Ni-O bonds) and the chemisorbed hydroxyl (O-H bonds) on the surface of NiO/ZnO heterostructures, respectively [37,38].

Figure 5 shows the UV-visible absorption spectra of ZnO NRs, NiO NSs, and NiO/ZnO heterostructures at room temperature. The band gap is calculated using Equation (1) [39]:(1)$$\alpha h\upsilon =C{\left(h\upsilon -E_{g}\right)}^{1/2}$$
where α is the absorption coefficient, C is a constant, and E_g_ is the band gap of the semiconductors.

The band gap energy absorption edges of ZnO NRs and NiO NSs are 385 nm and 317 nm, which correspond to their band gaps of 3.20 eV and 3.91 eV, respectively. For the NiO/ZnO heterostructures, due to the coupling of ZnO and NiO, more electrons are freely transferred from NiO with a higher Fermi level to ZnO with a lower level, which promotes the separation of holes and electrons and then effective heterojunctions are formed. Therefore, the band gap of NiO/ZnO heterostructures decreases to 3.05 eV, and the absorption edge showed a red shift from 385 nm to 406 nm [40].

### 3.2. Gas Sensing Properties

The sensors were placed in a four-liter chamber to test their sensing performance, and the test bias voltage was fixed during the gas sensing process. To allow the sensors to work at different operation temperatures, a heating wire was placed inside the ceramic tube. The response (R) of the sensors was defined as the ratio of R_a_ to R_g_ (when the resistance is reduced after the sensor is exposed to the target gas) or R_g_ to R_a_ (when the resistance is increased after the sensor is contacted with the targeted gas), where R_a_ is the resistance of the sensor measured in air and R_g_ is the resistance of the sensor when it contacts the gas detected.

Current density-voltage characterization curves of pure ZnO NRs, pure NiO NSs, and NiO/ZnO heterostructures were measured at 160 °C, and the results are shown in Figure 6a. All the data exhibit Ohmic behavior, and current density-voltage characteristics display linear patterns.

To investigate the optimum working temperatures of the gas sensor, we conducted the sensing process from room temperature to 300 °C. Figure 6b shows the responses of the NiO/ZnO-heterostructures-based sensor when exposed to 1 ppm of H_2_S at different working temperatures. As temperature increased, the response increased first and then decreased. The response value (R = 5) reached a maximum at 160 °C. Therefore, we concluded that for NiO/ZnO-heterostructures-based sensors, the optimal temperature is ~160 °C.

Figure 7 shows the dynamic response curves of the sensors for detecting H_2_S at 160 °C. When the concentration of H_2_S is 20 ppm, the response of pure ZnO NRs-based sensor is 2.7, whereas the response of NiO/ZnO-heterostructures-based sensor shows an improved response of 21.3. When the concentration of H_2_S is reduced to 0.1 ppm, the NiO/ZnO sensor still has an obvious response of 1.9, whereas the pure ZnO NRs-based sensor has almost no response. Apart from this, the pure NiO NSs-based sensor was also tested at 160 °C in 20 ppm H_2_S, and the response was only ~1.4, as shown in Figure 7d. This indicates that the enhancement of sensor’s performance is mainly due to the formation of NiO/ZnO heterostructures.

Based on the literature search, the performance of the H_2_S gas sensor based on different semiconductor materials or composite materials with various nanostructures are listed in Table 1. The NiO/ZnO heterostructures synthesized in this work showed a good response for H_2_S at a lower temperature, and its limit of detection is as low as 0.1 ppm.

To study the sensors’ selectivity, the dynamic response curves of a NiO/ZnO sensor exposed to H_2_S and other gases (such as NH_3_, C_2_H_6_O, NO_2_, and CO) for the same concentration of 20 ppm were measured at 160 °C. The results are shown in Figure 8a. For the NiO/ZnO-heterostructures-based sensor, if compared with the responses of 21.3 obtained when testing with H_2_S, the responses of several other gases were negligible. Therefore, the NiO/ZnO-heterostructures-based sensor has an excellent selectivity toward H_2_S.

To verify the stability and reproducibility, the NiO/ZnO-heterostructures-based sensor was tested repeatedly over a period of 25 days. The results shown in Figure 8b prove that it was stable to 1 ppm of H_2_S at 160 °C, with a maximum variation of about 6.9%.

### 3.3. Sensing Mechanism of NiO/ZnO-Heterostructures-Based Sensor

For the ZnO NRs-based semiconductor sensor, the surface-absorbed oxygen ions play a key role in its sensing mechanism [45,46,47]. When the ZnO NRs-based sensor is in contact with air, oxygen molecules in the atmosphere are absorbed onto the surface of nanorods and then quickly ionize into O_2_^−^ and O^−^ species by removing electrons near the surface of these ZnO nanorods, which leads to the formation of a depletion layer near their surface (Figure 9a). Generally, the O_2_^−^ species primarily react with target gas molecules when the temperature is below 150 °C, whereas the O^−^ species are dominantly involved in the reactions between 150 °C and 400 °C. Therefore, at 160 °C, the O ions on the surface of nanorods becomes dominant in the reaction [12,36]. Compared with O_2_^−^, O^−^ is more active and sensitive to a reducing gas. When H_2_S gas is released from the chamber and sensor’s surface, H_2_S molecules are oxidized by the O_2_^−^ and O^−^ ions on the surface of sensor and thus the electrons are released:(2)$$2H_{2}S+3O_{2}^{-}=2H_{2}O+2{\mathrm{SO}}_{2}+3e^{-}$$
(3)$$H_{2}S+3O^{-}=H_{2}O+{\mathrm{SO}}_{2}+3e^{-}$$

The released electrons increase the concentration of the majority carrier (electrons) and reduce the thickness of the depletion layer at the surface of ZnO NRs. All this contributes to the reduction in resistance and the increased current of the sensor, as shown in Figure 9a.

As the surface of ZnO NRs are modified by the NiO NSs, their responses to H_2_S are enhanced significantly. The reasons for this can be explained as follows: (1) The network-like structure increases the adsorption areas of oxygen ions. (2) The low valence Ni^2+^ can be oxidized easily to the high valence Ni^3+^ [13,48,49]; thus, oxygen ions are quickly adsorbed onto the sensor’s surface. (3) Due to the good contact between NiO and ZnO, a concentration difference is generated between holes and electrons at their interfaces. This concentration difference causes carrier diffusion, thus causing the energy band to bend in the depletion layer. When the carrier diffusion achieves an equilibrium condition, a space charge region is formed at their interfaces and a uniform Fermi level (EF) was obtained, as shown in Figure 10.

When the NiO/ZnO-heterostructures-based sensor is in contact with H_2_S gas molecules at 160 °C, H_2_S is mainly oxidized by the oxygen ions absorbed on the surface of ZnO NRs, releasing electrons into ZnO and increasing the concentration of the majority carrier (electrons). Therefore, the resistance of the sensor is reduced. For the semiconductor material, the holes decrease with the increase in electrons since $n_{0}\times p_{0}=n_{i}^{2}$, where $n_{0}$ is the electron concentration, $p_{0}$ is the holes’ concentration, and $n_{i}$ is the intrinsic carrier concentration. Therefore, the increase in electrons enhances the fast transport of electrons across the p-n junction. The electrons move to recombine with the holes, thus reducing the thickness of the depletion layer at the interfaces between NiO and ZnO [13,14]. Therefore, the resistance of sensor decreases and its conductivity increases. The schematic model for the sensing mechanism is shown in Figure 9b.

Figure 11 shows the resistance variation of NiO/ZnO-heterostructures-based sensor to 1 ppm H_2_S at 160 °C and 240 °C. Note that the resistance variation of NiO/ZnO-heterostructures-based sensor to 1 ppm H_2_S at 160 °C and 240 °C shows a completely opposite trend of dynamic responses at the lower and higher temperatures, with the critical boundary temperature being 160 °C.

When the temperature is higher than 160 °C, H_2_S mainly reacts with the oxygen ions (O^−^) on the surface of NiO NSs [50], thus releasing electrons into the NiO. Electrons recombine with holes, thus leading to a reduction in the majority carriers (holes) of NiO. The decrease in holes restrains the movement of electrons on both sides of the p-n junction. Therefore, the conductivity of the sensor is reduced. Besides, the dissociation of the reducing gas (e.g., H_2_S) causes the formation of crystal defects and deteriorates the conductive three-dimensional network structure. Consequently, the resistance of the sensor is increased. At higher temperatures, sulfur atoms are also decomposed from H_2_S molecules, which are then diffused into ZnO NRs and form ZnS [4,51]. Therefore, random structures of the n-p-n-type junction can be formed at the interfaces among ZnS, NiO, and ZnO, which leads to the increased barrier energy and also the sensor’s resistance.

## 4. Conclusions

A H_2_S semiconductor gas sensor based on NiO/ZnO heterostructures was fabricated and tested and its key sensing mechanisms were investigated. Comparing with the pure ZnO NRs, the sensing performance of the NiO/ZnO-heterostructures-based sensor was improved significantly within the concentration range of 0.1 to 20 ppm. The sensor has very good selectivity toward H_2_S compared to NH_3_, C_2_H_6_O, NO_2_, and CO, which can be attributed to the formation of p-n junction effects at the interfaces between NiO and ZnO. When the sensor is exposed to H_2_S at temperatures above 160 °C, the performance of the NiO/ZnO-heterostructures-based sensor changes from n-type semiconductor to p-type, and a sensing mechanism based on the p-n junctions was proposed.

## Figures and Tables

**Figure 1 nanomaterials-09-00900-f001:**
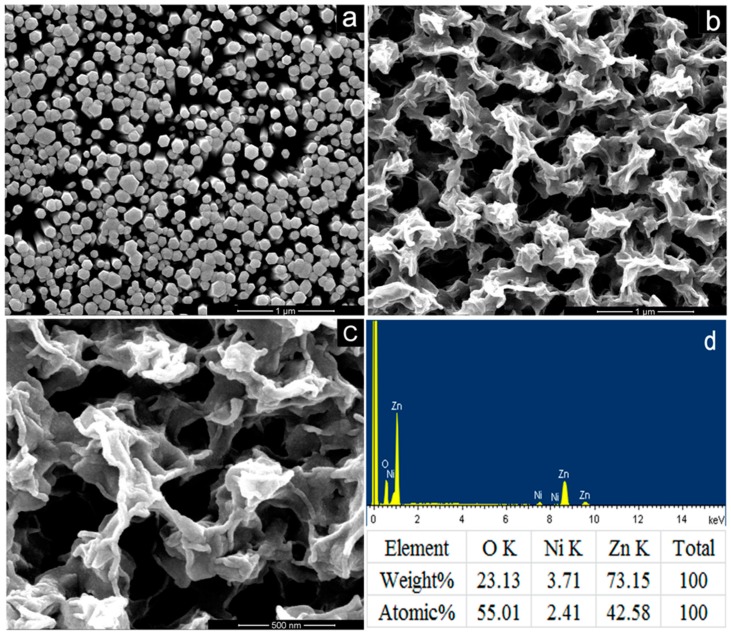
SEM images of (**a**) pure ZnO NRs, (**b**,**c**) NiO/ZnO heterostructures and (**d**) EDS of NiO/ZnO heterostructures.

**Figure 2 nanomaterials-09-00900-f002:**
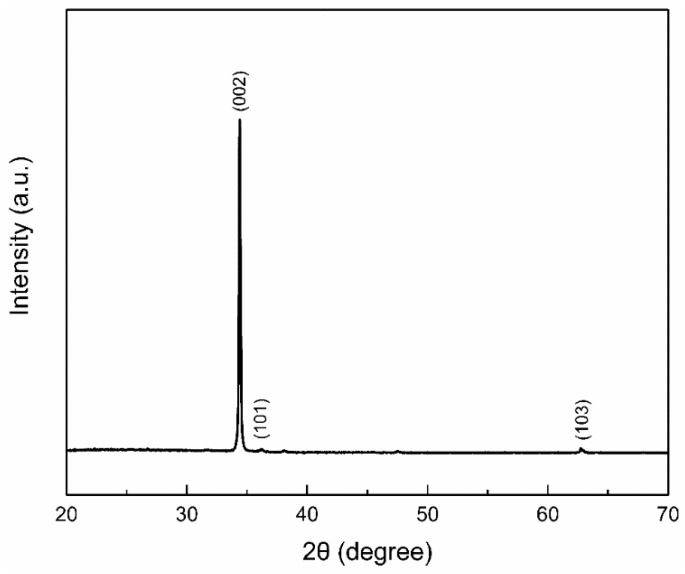
XRD patterns of ZnO NRs.

**Figure 3 nanomaterials-09-00900-f003:**
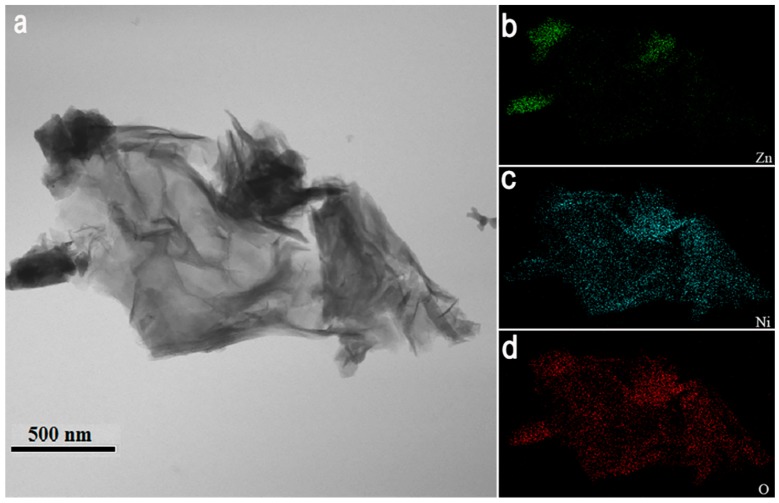
(**a**) TEM bright-field image of NiO/ZnO heterostructures and (**b**–**d**) elemental mapping of NiO/ZnO heterostructures.

**Figure 4 nanomaterials-09-00900-f004:**
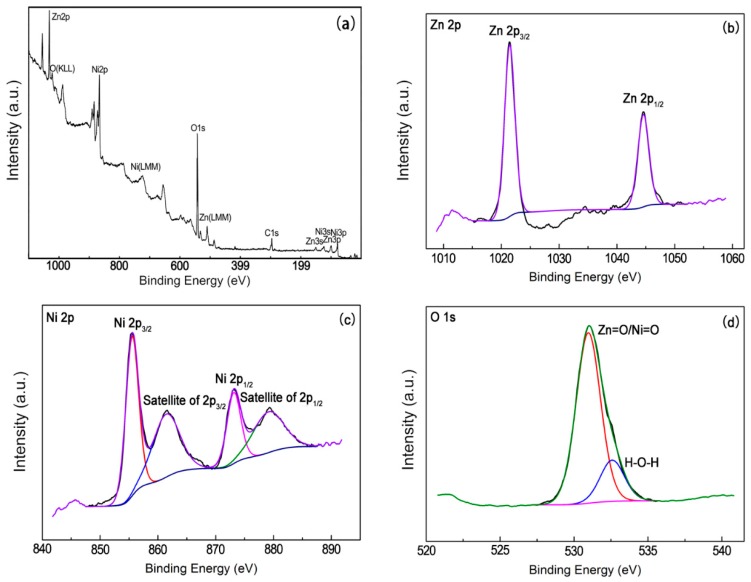
XPS spectra of (**a**) full spectrum, (**b**) Zn 2p, (**c**) Ni 2p, and (**d**) O 1s.

**Figure 5 nanomaterials-09-00900-f005:**
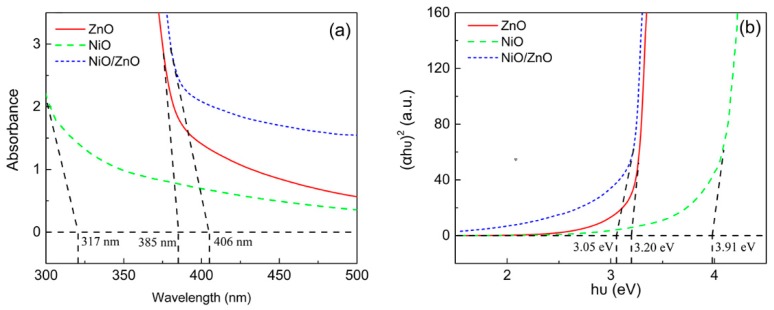
(**a**) UV-visible absorption spectra and (**b**) plot of (αhυ)^2^ vs. hυ of ZnO NRs, NiO NSs, and NiO/ZnO heterostructures.

**Figure 6 nanomaterials-09-00900-f006:**
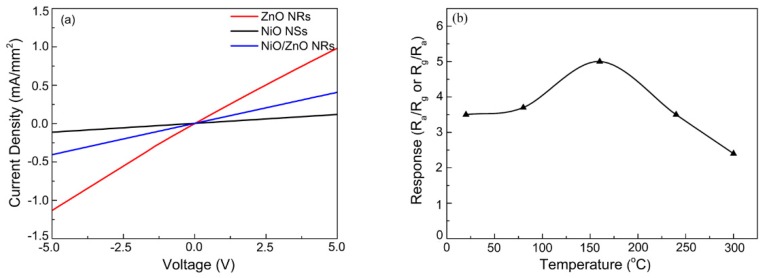
(**a**) Current density-voltage characteristics at 160 °C and (**b**) responses of NiO/ZnO-heterostructures-based sensor to 1 ppm of H_2_S at different operating temperatures.

**Figure 7 nanomaterials-09-00900-f007:**
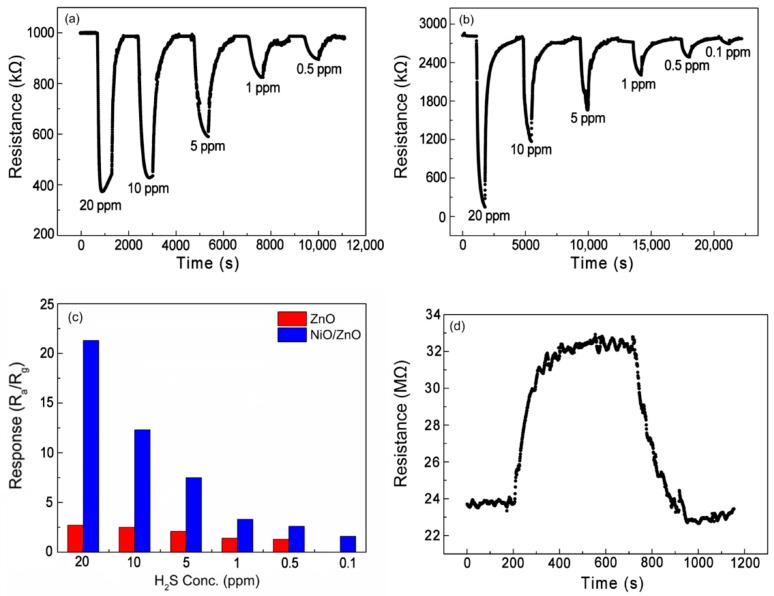
Dynamic response curves upon exposure to H_2_S with different concentrations at 160 °C exhibited by sensors based on: (**a**) pure ZnO NRs and (**b**) NiO/ZnO, and (**c**) responses histogram of sensors and (**d**) pure NiO NSs.

**Figure 8 nanomaterials-09-00900-f008:**
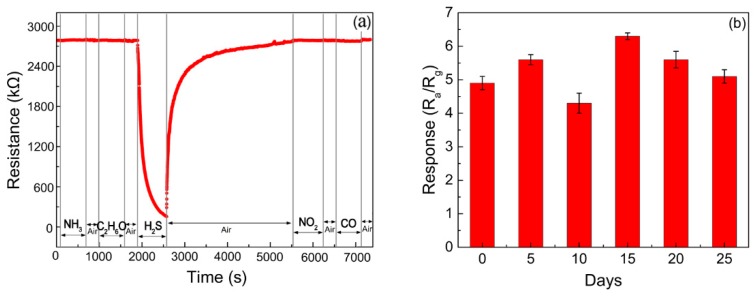
(**a**) Dynamic response curves of NiO/ZnO-heterostructures-based sensor to H_2_S and other gases (NH_3_, C_2_H_6_O, NO_2_, and CO) for the same concentration of 20 ppm was measured at 160 °C; (**b**) Stability of NiO/ZnO-heterostructures-based sensor to 1 ppm H_2_S at 160 °C.

**Figure 9 nanomaterials-09-00900-f009:**
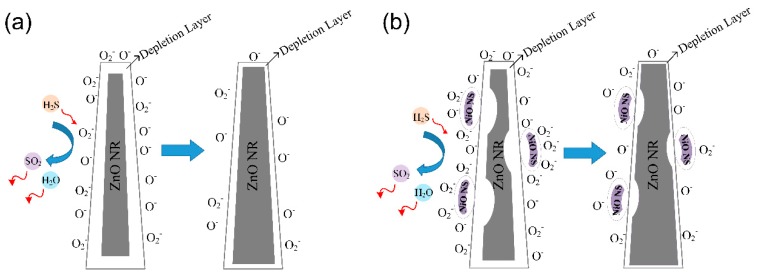
Schematic model for sensors exposed to H_2_S: (**a**) pure ZnO and (**b**) NiO/ZnO heterostructures.

**Figure 10 nanomaterials-09-00900-f010:**
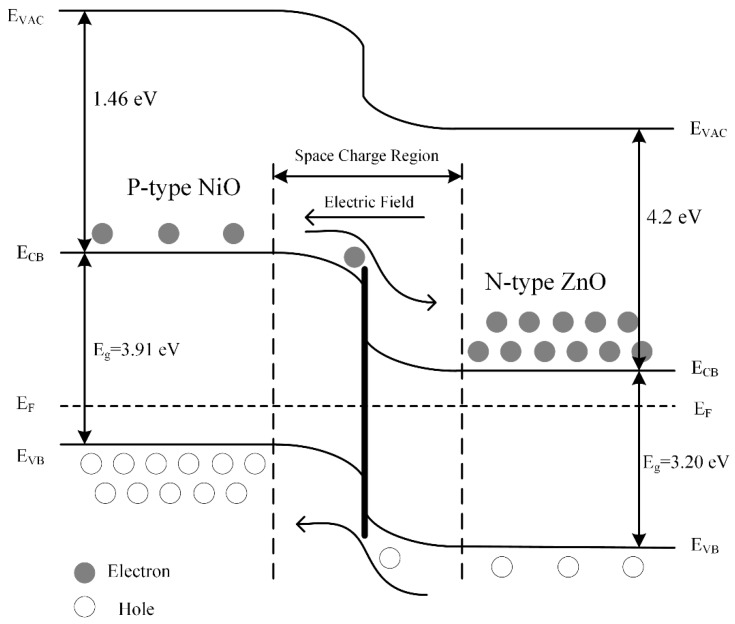
Energy band structure of p-NiO/n-ZnO heterostructures.

**Figure 11 nanomaterials-09-00900-f011:**
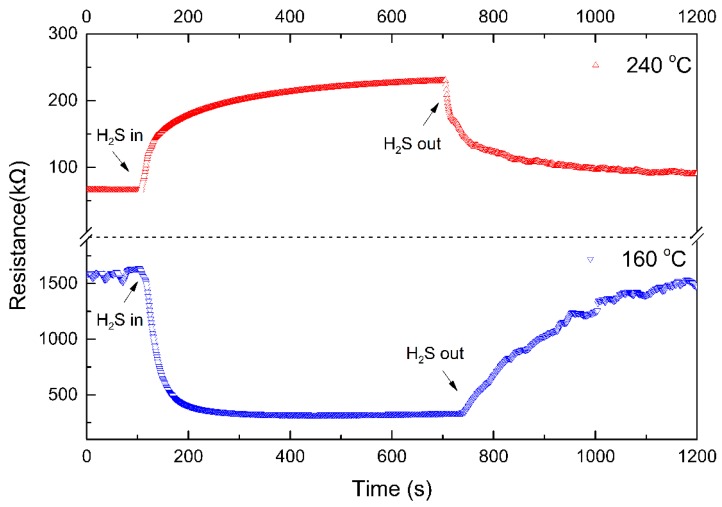
Dynamic response curves of NiO/ZnO-heterostructures-based sensor to 1 ppm H_2_S at 160 °C and 240 °C.

**Table 1 nanomaterials-09-00900-t001:** Comparison of the H_2_S gas sensors based on various semiconductors.

Material	Detection Limit (ppm)	Gas Conc. (ppm)	T (°C)	Response (R_a_/R_g_ or R_g_/R_a_)	Reference
ZnO Nanorods	-	100	100	23	[26]
CuO-ZnO Nanorods	-	100	100	39	[41]
NiO Thin film	1	50	92	20.6	[25]
Au-NiO Yolk-shell	1.25	5	400	44.2	[42]
CuO-NiO Microspheres	10	100	260	47.6	[43]
CuO-ZnSnO_3_ Nanowires	25	100	100	2.2	[44]
NiO/ZnO Heterostructures	0.1	20	160	21.3	This work

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
