# Peer review of "Heterostructured NiO/ZnO Nanorod Arrays with Significantly Enhanced H2S Sensing Performance"

_nanomaterials, 2019, doi:10.3390/nano9060900_

Round 1
Reviewer 1 Report
Many interesting results are provided in this article. The unique heterostructures of NiO/ZnO were successfully fabricated. Using Figures 9 and 10, the author explains the high response of the heterostructure against H2S gas clearly. Therefore, it is thought that this article is worth being placed in the Nanomaterials. However, the relation between the electrical properties appeared in Figure 6 (a) and the sensitivity against H2S gas is not clear. The author should explain the following items and submit the revised article.
1. The resistivity of the pure NiO NSs is high. Why the resistivity of NiO NSs falls by depositing onto the surface of ZnO NRs? Why the current I is increased linearly by increasing applied voltage V?
2. About the I-V characteristic of pure ZnO NRs, small knicks are observed at around plus 1.25 V and minus 1.25 V. Do they indicate a semiconducting property of ZnO NRs?
Author Response
Point 1: Many interesting results are provided in this article. The unique heterostructures of NiO/ZnO were successfully fabricated. Using Figures 9 and 10, the author explains the high response of the heterostructure against H2S gas clearly. Therefore, it is thought that this article is worth being placed in the Nanomaterials. However, the relation between the electrical properties appeared in Figure 6 (a) and the sensitivity against H2S gas is not clear. The author should explain the following items and submit the revised article.
Response 1: The electrical properties appeared in Figure 6 (a) indicated that the sensing materials were quasi-Ohmic behaviour. Their resistances won’t change with voltage, so no matter what the operating voltage is set, the test results won’t be affected.
Point 2: The resistivity of the pure NiO NSs is high. Why the resistivity of NiO NSs falls by depositing onto the surface of ZnO NRs? Why the current I is increased linearly by increasing applied voltage V?
Response 2: (1) For the resistance of NiO/ZnO heterostructures, it can be equivalent to NiO in parallel with ZnO. Therefore, the resistance of NiO/ZnO heterostructures is lower than pure NiO NSs. In addition, the resistance of NiO/ZnO heterostructures is slightly higher than pure ZnO, which may be a result of the extended depletion region near the NiO/ZnO junctions.
(2) Current I is increased linearly by increasing applied voltage V because it exhibits quasi-Ohmic behavior.
Point 3: About the I-V characteristic of pure ZnO NRs, small knicks are observed at around plus 1.25 V and minus 1.25 V. Do they indicate a semiconducting property of ZnO NRs?
Response 3: It is a problem with the test equipment. I have re-measured it several times and updated the data. Thank you.

Reviewer 2 Report
The manuscript entitled: "Excellent sensing performance for H2S gas of NiO/ZnO heterostructured nanorod arrays and mechanism study" proposed a H2S gas sensor with an amazing detection limit of 0.1 ppm.
Nevertheless some aspects are to be improved.
-The keywords have to be more specific "H2S gas sensor"
-Correct electricity properties with -electric properties.
-In Introduction specify the limits of H2S concentrations range that are dangerous for health.
-Introduce a TABLE to better show the used materials for H2S sensing, their performances in detection, the used temperatures and the references. The actual description is not appropriate and sometimes is confusing or difficult to assess. Limit to the materials tested for H2S sensing and complete the literature with actual one.
-Your References have to be actualized with 2016-2019 published papers.
-The synthesis has to be clearly presented and the number of moles have to be calculated for all syntheses.
- Punctuation is not correct in this section. Some sentences begin with "And" and it is not OK.
-Explain clearly why is the second part using nitrates necessary, after using acetates.
-Section 2.2 is confusing. Is it everything deposited on Al2O3? The same for section 3.1. What is the meaning of 2.41% Ni content? What amount was estimated? What was the molar ratios?
- Correct : p-n heterostructures on the interfaces with
p-n heterostructures at the interfaces.
-XRD in this case does not bring any proof.
- Figure 3, suggest another distribution than that one described. Ni and O seems to be overlapped and Zn is scarcely located. The Figures are to be larger.
-Gave a reference for equation (1). Describe each parameter. Figure 5 is not visible.
-Units have to be added to all measurements:
"When the concentration of H2S gas is 20 ppm, 169 the response of pure ZnO NRs based sensor is 2.7 ????"
- ALL figures are inappropriate.
-The mechanism part contains many repetitions and must be reformulated in a clear and logical way. Are in this particular case the conditions for Ni to transfer one electron from s to d layer? Can you explain this?
- Conclusion section is not appropriate and the last sentence is wrongly written.
Author Response
Point 1: 1. The keywords have to be more specific "H2S gas sensor".
Response 1: The keywords have update.
Point 2: Correct electricity properties with -electric properties.
Response 2: It has been corrected.
Point 3: In Introduction specify the limits of H2S concentrations range that are dangerous for health.
Response 3: It has been added to the introduction.
Point 4: Introduce a TABLE to better show the used materials for H2S sensing, their performances in detection, the used temperatures and the references. The actual description is not appropriate and sometimes is confusing or difficult to assess. Limit to the materials tested for H2S sensing and complete the literature with actual one.
Response 4: A table has been added.
Point 5: Your References have to be actualized with 2016-2019 published papers.
Response 5: I have updated some of the references.
Point 6: The synthesis has to be clearly presented and the number of moles have to be calculated for all syntheses.
Response 6: The molecular mass of chemicals has been calculated and updated.
Point 7: Punctuation is not correct in this section. Some sentences begin with "And" and it is not OK.
Response 7: The grammatical and punctuation mistakes had been revised.
Point 8: Explain clearly why is the second part using nitrates necessary, after using acetates.
Response 8: The method of classic literatures is like this, and I followed them.
Point 9: Section 2.2 is confusing. Is it everything deposited on Al2O3? The same for section 3.1. What is the meaning of 2.41% Ni content? What amount was estimated? What was the molar ratios?
Response 9: ZnO NRs are grown on Al2O3 and NiO NSs are deposited on ZnO NRs. Therefore, NiO NSs and ZnO NRs are on Al2O3. Ni content comes from the measurement of EDS, which is not accurate. EDS proves the existence of NiO.
Point 10: Correct: p-n heterostructures on the interfaces with p-n heterostructures at the interfaces.
Response 10: It has been modified to “at the interfaces”.
Point 11: XRD in this case does not bring any proof.
Response 11: XRD can’t directly explain the existence of NiO for the content of NiO is too small, while it confirmed the existence and directional growth of ZnO NRs.
Point 12: Figure 3, suggest another distribution than that one described. Ni and O seems to be overlapped and Zn is scarcely located. The Figures are to be larger.
Response 12: The sample used in the TEM was scraped from the ceramic tube, so the morphology was not exactly the same as the SEM image. Since the light parts of figure 3a is NiO, and the atomic ratio of Ni to O is 1:1, the elemental mappings of Ni and O seem to overlap. Actually, due to the presence of ZnO (the dark parts in figure 3a), the number of O atoms is more than the number of Ni atoms corresponding to the position of Zn.
Point 13: Gave a reference for equation (1). Describe each parameter. Figure 5 is not visible.
Response 13: The reference has been gave. All figures are updated.
Point 14: Units have to be added to all measurements: "When the concentration of H2S gas is 20 ppm, 169 the response of pure ZnO NRs based sensor is 2.7 ????"
Response 14: The ratio of Ra to Rg was defined as the response (R), where Ra is the resistance in air and Rg is the resistance in target gas. The unit of R is 1. It is mentioned in section 3.2.
Point 15: ALL figures are inappropriate.
Response 15: All the figures are updated.
Point 16: The mechanism part contains many repetitions and must be reformulated in a clear and logical way. Are in this particular case the conditions for Ni to transfer one electron from s to d layer? Can you explain this?
Response 16: It has been reformulated. Because the focus of this article is on the impact of pn junction on sensor performance, the second question you give is a good research direction. Since I know less about this, I can’t give a reasonable explanation now, but it can be studied as a follow-up work. Thank you.
Point 17: Conclusion section is not appropriate and the last sentence is wrongly written.
Response 17: It has been altered.

Round 2
Reviewer 1 Report
I confirmed that the referee's comments are reflected and the article is improved.
Author Response
Thanks a lot for your attention and careful reviews on our manuscript nanomaterials-504177.
According to your suggestions, we have made a careful revision on the previous version of the manuscript, and these changes of English language and style will not influence the content and framework of this paper. Changes has been listed in a document named “modification list”, and revised portion are marked in yellow (rewritten) or green (added) in revised paper.
I hope that the correction will meet with approval. Thank you.

Reviewer 2 Report
The manuscript was modified but not significantly improved, so that many formulations that are not clear are still present:
In the Abstract this phrase is completely confusing:
1. "Compared to the pure ZnO nanorod arrays, the netlike structures of NiO/ZnO heterostructures increased the area of surface to adsorb oxide ions."
2. This sentence has no meaning: " Which was ascribed to the heterostructures constituted by ZnO and NiO."
3. "well chemical stability" is not an information.
4. "In these metal oxides, ZnO nano-material is the most representative one." Please correct with
Among these....
5. Sentence lacks to be clear and sound:" modified with 50 Cu to 10 ppm H2S is 18.7 at 230 oC, while detection limit is 1 ppm. "
6. Figure 2b has no meaning. No change was evidenced, so that the text and only Fig 2a is enough.
7. What is this meaning? please explain: "Besides, two satellite 124 peaks located at the binding energies of 861.5 eV and 879.3 eV are from Ni 2p3/2 and Ni 2p1/2, 125 respectively[16]."
8. The meaning of "I-V characterization curves" is inappropriate. Please describe it to be understandable. You cannot refer to the density of current (intensity) or to the potential in this way.
9. What do you mean by RESPONSE???? Describe it clearly. All the paragraph has to be reformulated " even if it is a ratio. Clarify also Figure 7, put in the brackets the meaning.
When 164 the concentration of H2S is 20 ppm, the response of pure ZnO NRs based sensor is 2.7, while 165 the response of NiO/ZnO heterostructures based sensor improves almost to 21.3. When the concentration of H2S is reduced to 0.1 ppm, the NiO/ZnO heterostructures based sensor still has an obvious response of 1.9, while the pure ZnO NRs based sensor has almost no response."
10. Table 1 represents a comparison of your results with published data. Correct as
Comparison of the H2S gas sensors BASED ON various semiconductorS .
11. The Stability is not relevant. The errors are very high in both directions, so that an error bar is needed to prove the behavior.
12. What is the provenience of O- species? Describe in detail.
13. Please provide a proof for the formation of Nickel subsulfide. and establish a clear mechanism, because too many oxidation reactions are involved.
14. The Conclusions are different than the statements from discussions. Please reduce the arguments to only sound scientifically determined and unify them in the different chapters.
Author Response
Thank you for your attention and careful reviews on our manuscript nanomaterials-504177. Those comments are all valuable and very helpful for revising and improving our paper. According to your detailed suggestions, we have made a careful revision on the previous version of the manuscript. In addition, some changes of English language and style have been finished and which will not influence the content and framework of this paper. Changes has been listed in another document named “modification list”, and revised portion are marked in yellow (rewritten) or green (added) in revised paper.

Round 3
Reviewer 2 Report
I am very glad to see a major improvement of the manuscript "Heterostructured NiO/ZnO nanorod arrays with significantly enhanced H2S sensing performance ".
There is still one simple request.
Please Complete in Figure 6, I-V with density of current or current intensity and potential, because in actual form it is not the best way for readers.
In conclusion, I recommend its publication, after this change.